# GENEVAL: A FRAMEWORK TO EVALUATE FEASIBILITY OF DOMAIN GENERALIZATION

## ABSTRACT

This paper proposes a novel methodology for evaluating whether a learned hypothesis is generalizable to a new domain. GenEval extracts underlying models that represent effects of causal factors on domain data and labels and uses a novel model divergence detection mechanism based on conformal inference to evaluate significant shift in causal factors in the new domain. As such, GenEval can predict the performance of a learned hypothesis in the new domain without the need for execution in the new domain. We evaluate GenEval on single, multi-dimensional time series applications as well as challenging medical imaging case studies on diabetic retinopathy in both single and multi-domain generalization experiments.

## 1 INTRODUCTION

This paper proposes, *GenEval*, a novel technique for evaluating necessary conditions for domain generalization (DG). DG can be achieved under two assumptions Vuong et al. (2025); Cha et al. (2021); Arpit et al. (2022): a) **label identifiability**, which implies that different causal factors cannot result in the same data if the causal factors cause a label distribution shift, and b) **causal support**, which implies that the combination of source domains encompass all causal factors that cause distribution shift. Based on these assumptions two necessary conditions for DG have been proven in Vuong et al. (2025); Cha et al. (2021); Arpit et al. (2022):

*NC1: Optimal hypothesis for training domains,* which requires the learning hypothesis to minimize loss function in each training domain. NC1 is satisfied through empirical risk minimization (ERM).

*NC2: Invariance preserving representation function,* the learning hypothesis should accurately represent the effects of each causal factor on the data distribution across domains. To the best of our knowledge, there is no technique that is proven to satisfy NC2.

Several techniques have been proposed for DG Vuong et al. (2025); Cha et al. (2021); Arpit et al. (2022), which mainly focus on satisfying NC1 through ERM and propose several heuristics at satisfying NC2 and have performance improvements with respect to the ERM baseline Arpit et al. (2022). However, two major questions still remain unanswered:

*Q1: How do we ascertain that a target domain satisfies causal support assumption?*

*Q2: How do we ascertain that a learned hypothesis indeed satisfies invariance preservation condition?*

To answer Q1, *GenEval*, utilizes a combination of Koopman operator guided sparse model recovery and model conformal inference that identifies the most relevant subset of causal factors affecting data distribution across source domains and establishes *domain conformal boundaries*. Causal factors derived from target domains can then be checked against the conformal boundaries to determine significant deviation indicative of violation of causal support assumption.

Q2 is answered by applying the same combination of sparse model recovery and model conformal inference on the representation functions of a learned hypothesis across source domains to establish *hypothesis conformal boundaries*. The outputs of the representation functions of the learned hypothesis on target domain can be cheked against the hypothesis conformal boundaries to determine significant change indicative of violation of invariance preservation condition.

The architecture of *GenEval* is guided by a novel exploration of the Mori-Zwanzig (MZ) formulation Lin et al. (2021); Venturi & Li (2023) for nonlinear functional representation to overcome steady state assumptions through explicit modeling of external input effects. This enables *GenEval* to automatically explore and include new causal factors in different source domains and update the domain conformal boundaries.

We evaluate the effectiveness of *GenEval* in identifying domains that do not satisfy causal support, and hypothesis that do not satisfy invariance preservation in 14 time domain regression analysis

and two image domain supervised classification case studies including challenging applications of detection of seizure onset zone from functional resting state magentic resonance imaging and classification of diabetes retinopathy stages from fundus imaging. *GenEval* is evaluated in both single and multi-source domain generalization problems and shows good accuracy in predicting performance of state-of-the-art baseline DG techniques on benchmark datasets. In addition, we show that when causal factors are explicitly modeled in a learning hypothesis, it can outperform SOTA benchmarks on real world datasets for both time series and imaging case studies.

## 2 PRELIMINARIES

We introduce some basic definitions and problem statements related to DG in this section.

### 2.1 STANDARD DOMAIN GENERALIZATION SETTING

For generality, $X \in \mathcal{X}$ denotes the input space and $Y \in \mathcal{Y}$ the target space; for example,

$$\mathcal{X} = \begin{cases} \mathcal{R}^{T \times P} & \text{for multivariate time series of length } T \text{ with } P \text{ channels,} \\ \mathcal{R}^{N \times M \times 3} & \text{for images of height } N, \text{ width } M, \text{ and 3 colors,} \end{cases}$$

and

$$\mathcal{Y} = \begin{cases} \mathbb{R}^{T' \times P} & \text{for forecasting the next } T' \text{ time steps,} \\ \{1, \ldots, C\} & \text{for } C\text{-way image classification.} \end{cases}$$

We consider set $\mathcal{D}$ to represent a set of domains $D \in \mathcal{D}$. We define $Y_\delta$ as a set of logits that determines the performance of any learning hypothesis in the learning task. For time series regression task $Y_\delta \subset \mathcal{R}^T$, whereas for supervised classification task $Y_\delta = \{\delta \in \mathcal{R}^C : \|\delta\|_1 = 1 \wedge \delta > 0\}$. The learning hypothesis is a function $h : X \rightarrow Y_\delta$. For time series regression task $h(x) = y'$, is a $T$-tuple, where $y'$ is an estimate of the ground truth $y$. For supervised classification task it predicts a $C$-tuple $h(x) = \{h(x)[1], \ldots h(x)[C]\}$ such that $h(x)[i] = p(y = i|x)$ is the probability that data $x$ is from class $i$. We define a loss function $l : Y_\delta \times Y \rightarrow \mathcal{R}$ which for time series regression can be root mean square error and for classification can be cross entropy. The loss for a given domain $D$ is $\mathcal{L}(h, D) = \mathbb{E}_{(x,y) \in D}[l(h(x), y)]$.

**Domain Generalization Problem Statement:** Given $K$ source domains $\{D_1 \ldots D_K\}$, the aim of DG is to learn a globally optimal hypothesis $h^*(X) = \bigcap_{D_J \in \mathcal{D}} \arg\min_h \mathcal{L}(h, D_J)$ that minimizes the loss in any domain $D_J \in \mathcal{D}$ even if $D_J \notin \{D_1 \ldots D_K\}$.

Note that two versions of the DG problem has been studied in recent literature: a) multi-source DG Wang et al. (2022a), where $K > 1$ and b) more challenging single-source DG, where $K = 1$ Galappaththige et al. (2024).

**Standard data generation process:** It is commonly assumed that in DG, $X, Y$ in a domain $D$ is generated through an underlying process that can be expressed using deterministic process models. The underlying data generation process for a domain $D$, can be expressed as a state space model with: a) a set of endogenous variables $V = \{X, Y, Z_c, Z_d, D\}$, where $Z_c$ expresses causal factors that affect labeling process to generate $Y$, while $Z_d$ encompasses domain specific factors that are correlated with $X$, and $D$ expresses domain specific factors, b) a set of exogenous inputs $U = \{U_x, U_y, U_c, U_e, U_D\}$ related to each variable in $V$, and c) a set of deterministic equations $\psi = \{\psi_x, \psi_y, \psi_c, \psi_d, \psi_D\}$ expressing the data generation process for each variable in $V$. The data generation process can be expressed as a series of non-linear deterministic functions as follows:

$$d = \psi_D(u_D), u_D \sim Pr(U_D), z_d = \psi_d(d, u_e), u_e \sim Pr(U_e), z_c = \psi_c(d, u_c), u_c \sim Pr(U_c), \tag{1}$$
$$x = \psi_x(z_c, z_d, u_x), u_x \sim Pr(U_x), \text{ and } y = \psi_y(z_c, u_y), u_y \sim Pr(U_y),$$

where $\sim$ denotes membership to a probability distribution $Pr(.)$.

**Composite hypothesis:** In most time series and image based applications hypothesis function $h$ very commonly takes a composite structure $h = f \circ g$, where $g : X \rightarrow S$, that maps the data to a latent space $S$ and $f : S \rightarrow Y_\delta$ that maps the latent space to the logits. In this paper, we restrict our hypothesis to composite structures that has widespread applicability.

**Differentiable process model:** The DG setting does not restrict the deterministic functions $\psi$ to be continuous and differentiable. However, in most practical applications the functions $\psi$ if not already continuous and differentiable can be approximated as such with high fidelity as seen in many time series Banerjee & Gupta (2024); Kaheman et al. (2020) and image based applications Garnung Menén-dez (2024); Chan et al. (2003). Moreover, this assumption is implicitly undertaken in most prediction schemes that use gradient descent as a learning strategy Arora et al. (2022). Nevertheless, even

if the functions $\psi$ are non-differentiable and discontinuous, we can still represent $\psi$ using hybrid differential dynamics as shown in several well established works David et al. (2011); Mathews & Carlson (2006); Slightly & Gadsen (1998); Bennett & Thompson (2005); Camacho & Bordes (2007); Smith & Zhang (2008) by increasing the number of state variables, where the new state variables allow for splitting the differential dynamics around the discontinuities resulting in piecewise models. Hence in this paper, we represent the data generation process as a high dimensional state space forced differential dynamics, for image partial differential dynamics (PDD), for time-series ordinary differential dynamics (ODD) with the generic form given by Eqn. 2.

$$\nabla V = \Psi(V, U), \text{ where } \Psi(.) = \{\partial\psi_x(.), \partial\psi_y(.), \partial\psi_c(.), \partial\psi_d(.), \partial\psi_d(.)\}^T, \tag{2}$$

where $\nabla$ and $\partial$ are differential operators for variables and functions respectively, either on scalar (time series) or vector space (image data).

**Label Identifiability:** The only observable variables in the DG problem are $X$ and $Y$, while $Z_c$, $Z_d$ and $D$ are not observable. However, for any hypothesis to accurately represent the causal $Z_c$, the components of function $\psi_y$ and $\psi_x$ has to be identifiable with only the measurements of $X$ and $Y$. The concept of identifiability has been extensively used in control theory Grewal & Glover (2003) and model recovery Banerjee & Gupta (2024) literature. It implies that two causal factors $z_c, z_c' \in Z$ cannot affect labels $Y$ or $Y_\delta$ without affecting data $X$. Mathematically, it implies $\frac{\partial Y_\delta}{\partial X} = \frac{\partial Y_\delta}{\partial z_c}\frac{\partial z_c}{\partial X}$ is non-zero and well-defined for any causal factor $z_c$.

**Invariant representation function:** The assumption of label identifiability directly implies that there is a set of deterministic functions $G_c$ that can accurately model the relationship between $z_c \rightarrow Y$ and $z_c \rightarrow X$ which can be extracted from the measurements of $X$ and $Y$ such that $\forall g_c \in G_c, Pr(Y|g_c(X)) = Pr(Y|Z = z_c)$.

**Causal Support:** Union of all domains $\bigcup\limits_{i=1}^{K} D_J$ encompasses all possible causal factors in $Z_c$.

**Sufficient Conditions for DG:** Traditionally, hypothesis search for DG problem has focused on achieving a set of *sufficient conditions* for DG, i.e., if a hypothesis generalizes to target domains then $h = f \circ g$ should satisfy the following properties Vuong et al. (2025):

- $g$ is a member of a set $G_c$ of ***invariant representation function***.

- The set of source domains used for training $D_1 \ldots D_K$ have ***causal support*** and also *covers the domain specific factor space*, $Z_c \times Z_d$.

**Necessary conditions for DG:** If the learned hypothesis $h = f \circ g$ satisfies the following two necessary conditions (as identified in recent literature Vuong et al. (2025)) then it is a globally optimal hypothesis :

- *NC1, Optimal hypothesis for training domains:* The learned hypothesis $h = f \circ g$ should minimize the loss function in all source domains.

- *NC2, Invariance preservation:* The learned representation $g$ should preserve all mutual information content between $X$ and the invariant representation functions $g_c \in G_c$, i.e., $I(g(x), g_c(X)) = I(X, g_c(X))$, where $I(.,.)$ is a function that measures mutual information content such as $\mathcal{H}$-divergence or Hellinger or Wasserstien distance.

### 2.2 REVIEW OF SOTA DG AND CHALLENGES

DG approaches can be categorized into four types:

**Representation alignment:** These approaches assume the domain adaptation setting where there is access to the target domain dataset Ben-David et al. (2010); Ben-Hur et al. (2001); Phung et al. (2021); Zhou et al. (2020a); Johansson et al. (2019). The fundamental aim is to satisfy the invariance preservation necessary condition by explicitly incorporating divergence of source and target domain representations in the loss function (domain loss) and learning consistent representations across source and target domain. While this is theoretically sound approach, it cannot be applied to DG since target domain data is not available.

**Invariant prediction:** These approaches Arjovsky et al. (2020); Ahuja et al. (2020); Krueger et al. (2021); Li et al. (2022); Mitrovic et al. (2020); Zhang et al. (2023) focus on satisfying the sufficient conditions of DG and use causal learning methods to learn a representation function that is potentially a member of the invariant representation function set $G_c$. However, there are two major drawbacks of these techniques: a) while they achieve NC1, they ignore NC2, and b) there is no analysis on whether a target domain $D_T$ satisfies causal support and covers domain specific factor space $Z_c \times Z_d$, instead it is just assumed.

**Data augmentation:** There are two classes of approaches:

a) methods Mitrovic et al. (2020); Wang et al. (2022b); Zhou et al. (2020a; 2021); Zhang et al. (2017); Wang et al. (2020b); Zhao et al. (2020); Yao et al. (2022a); Carlucci et al. (2019); Yao et al. (2022b) that augment training datasets by incorporating learnable transformations on the original samples thereby improving the probability of the learned hypothesis $h$ to learn a representation function $g$ that conforms to a member of $G_c$ (first sufficient condition).

b) methods Kamboj et al. (2025) that incorporate new causal factors and generate new data based on a convolution of existing training datasets and the effect of the causal factors on the training data. These techniques focus on satisfying both the sufficient conditions by improving probability of learning a invariant representation function and improving causal support.

**Ensemble Learning:** These approaches Zhou et al. (2021); Ding & Fu (2017); Wang et al. (2020a); Mancini et al. (2018); Cha et al. (2021); Arpit et al. (2022) combine the source domains into a single dataset and train multiple hypothesis with the same architecture with different splits of the combined training data. During prediction they pass target domain data into each learned hypothesis and combine the outputs of each hypothesis. Alternatively, these approaches may also perform weight averaging resulting in reduction of divergence across predictions from each hypothesis. Recently, seminal work by Vuong et al. (2025) has theoretically shown that ensemble learning approaches achieve the best performance among all other DG approaches since they satisfy both the NC1 and NC2 conditions.

**Challenges of SOTA DG:** Despite performance improvements with ensemble learning and proposition of novel DG techniques based on necessary and sufficient conditions, DG performance on real world datasets have been abysmal with an average accuracy across 5 datasets in Domainbed benchmark as 63%. In critical medical applications of Diabetic Retinopathy (DR) stage detection, the performance on multi-source domain generalization of SOTA techniques is even more dismal as shown in Table 1. Such accuracies prevent application of any published research in clinical practice.

We posit that there are two fundamental reasons current DG methods show poor performance across domains: a) for a target domain the causal support property is only assumed and never checked resulting in selection of target domains that are infeasible to generalize, and b) invariance preservation property of a learned hypothesis is not validated in source domains resulting in learning hypotheses that are not invariance preserving.

Table 1: Comparison of model performance across DR datasets in the MDG setting. Train on three domains and test on the one in the table columns.

| Method | Aptos | Eyepacs | Messidor | Messidor 2 | Avg. |
|---|---|---|---|---|---|
| MMD Li et al (2018a) | 49.3±1.0 | 69.3±1.1 | 64.1±4.8 | 69.6±0.6 | 63.1 |
| CDANN Li et al. (2018b) | 48.1±0.7 | 73.1±0.3 | 55.8±1.8 | 61.2±1.3 | 59.5 |
| SD-ViT Sultana et al. (2022) | 46.5±0.8 | 71.1±0.7 | 63.9±0.9 | 71.4±0.2 | 63.2 |
| SPSD-ViT Jayanga et al. (2023) | 51.6±1.1 | 73.3±0.4 | 64.0±1.4 | **72.9±0.1** | 65.5 |
| ERM-ViT Vapnik (1999) | 48.5±0.9 | 70.7±1.7 | 62.7±1.6 | 69.5±2.5 | 62.9 |

### 2.3 PROBLEM STATEMENT

Given a set of $K$ source domains $\{D_1 \ldots D_K\}$ defined by $V$ and $\Psi$ in Eqn. 2 with causal factor set $Z_c$ and domain specific factor set $Z_d$ we solve two problems:

**Support violation detection:** For a target domain $D_J \notin \{D_1 \ldots D_K\}$ with causal factor set $Z_c^{(J)}$ and domain specific factor set $Z_d^{(J)}$, is $Z_c^{(J)} \subseteq Z_c$ and $Z_d^{(J)} \subseteq Z_d$?

**Invariance preservation violation:** For a hypothesis $h = f \circ g$ learned on source domains, is $I(g(X), g_c(X)) = I(X, g_c(X))$ in the target domain $D_J$?

**Significance:** Answers to these questions can help evaluate the feasibility of DG on a target domain by learning hypothesis from a set of source domains. This can also help select the domains that should be considered as source domains better DG. It can also be used to evaluate the generalization capability of a hypothesis beyond its accuracy on a limited number of target domains.

## 3 SOLUTION SKETCH

In this section, we will focus on solving the support violation detection problem. The solution for invariance preservation violation is simple extension.

**Support violation detection solution sketch:** We explain the intuition behind the solution and the solution sketch using a simple example of non-linear one-dimensional time series regression problem. Given a dataset $X \subset \mathcal{R}^{1000 \times 1}$, and the label set $Y \subset \mathcal{R}^{100}$, which is the next 100 samples of $X$. The task is to learn a hypothesis from $X$ that accurately predicts $Y$. The causal structure of $X$ is described by a causal factor $Z_c$, a domain specific factor $Z_d$ and an exogenous variable $u$. Assume

that the ground truth causal structure is given by the Lorenz oscillator system as in Eqn. 3.

$$Y = X, \; \frac{dX}{dt} = 10(Z_c - X) + u, \; \frac{dZ_c}{dt} = 28X - Z_c - XZ_d, \; \frac{dZ_d}{dt} = 2.67Z_d + XZ_c \qquad (3)$$

Consider two domains, $D_1$, and $D_2$ where the data $X$ and label $Y$ are generated by assuming $u = 10$, a constant input and $u = \sqrt{\frac{2}{K}} cos(\omega t + \theta)$. Hence these two domains differ significantly in the function $\psi_x$ that relates the causal factor $Z_c$ with $X$. The aim of the support violation detection is to flag domain $D_2$ as infeasible for DG.

If we have access to measurements of $Z_c$ and $Z_d$, then we can utilize equation discovery tools such as sparse identification of non-linear dynamics with model predictive control (SINDY-MPC) Kaheman et al. (2020); Kaiser et al. (2018) to extract the exact function set $\Psi$ that describes data from each domain $D_1$ and $D_2$. Fig. 1 shows the derived equations using data from the two domains. Here we see significant difference between the structures of the underlying causal structure of each domain, where in $D_1$ there is no effect of $Z_c$ on $X$ but in $D_2$ the original equation in Eqn. 3 could be derived with characterization of impact of the causal factor $Z_c$ on $X$. Hence, the solution to the support violation detection problem could have two steps: a) equation discovery, and b) significant causal structure change detection.

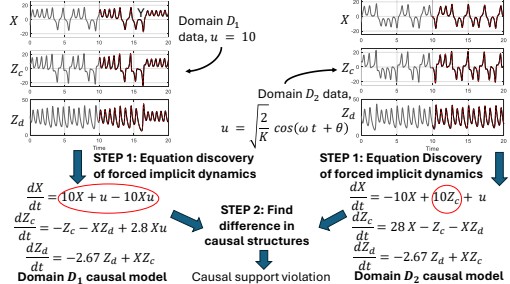

Figure 1: Solution sketch for solving support violation detection.

**Technical challenges:** Although the solution is intuitive, there exists several technical challenges.

*a) Unmeasured causal factors* - $Z_c$ and $Z_d$ are not measurable variables. Hence, the solution method requires **extraction of unmeasured causal factors in a nonlinear dynamical systems from data with forcing inputs**.

*Drawbacks of existing techniques:* There are two classes of equation discovery techniques: a) sparse regression following Koopman theory Chen et al. (2012), pioneered by Kaiser et al. (2018), where the non-linear regression is converted to a linear regression problem by creating a library of non-linear functions of the measurable state variables, and b) combination of neural architectures with sparse regression guided by the MZ extension of Koopman theory Venturi & Li (2023); Lin et al. (2021) for modeling unmeasured causal factors. Sparse regression techniques cannot recover unmeasured causal factors. Extensions such as incorporating differentials of state variables as new state vectors have been tried Fasel et al. (2022) but they can only improve reconstruction accuracy of measured signals but have been shown to be inaccurate in recovering the unmeasured causal factors Fasel et al. (2022). Moreover, such techniques do not recover dynamics under forcing inputs. Techniques following MZ extension of Koopman theory make an important assumption of steady state that precludes them from being capable of recovering dynamics under forcing inputs.

*Novel contribution: GenEval* takes a new look at MZ extension and proposes a novel equation recovery architecture that can **recover unmeasured causal factors in presence of forcing inputs**.

*b) unsupervised change detection under uncertainties* - Since the target domain data is noisy and previously unseen, we do not have a training dataset with labels on the causal structures. Hence, the **change in causal structure** has to be detected in an **unsupervised manner under uncertainties**.

*Drawbacks of existing techniques:* A change in causal structure can be described as a source for anomalies. There has been a plethora of works in anomaly detection both in time series Liu et al. (2024b); Liu & Paparrizos (2024); Kim et al. (2022) and image domain Liu et al. (2024a). While individual techniques are data modality specific, for this work we seek a technique that can operate across modalities. As such we require a change detection technique that can operate on causal structures rather than on time-series or image data. The fundamental operator for change detection in overwhelming majority of anomaly detection techniques is extreme value theory (EVT) Liu & Paparrizos (2024) for out of distribution (OOD) sample detection. EVT has underlying assumptions about the distribution. When applied on finite sample data, it is easy to verify whether the data matches the distribution assumption of EVT. However, when applied on implicit causal structures there is no way of apriori verifying satisfaction of EVT assumptions. Hence, application of EVT on causal structures may not result in accurate change detection.

*Novel contributions: GenEval* proposes a model free unsupervised change detection method based on statistical conformal inference that can **detect changes in implicit causal factors**.

## 4 THEORETICAL FOUNDATIONS OF *GenEval*

In this section, we discuss the theoretical foundations of unmeasured causal factors recovery and model free change detection finally leading us to *GenEval* technique.

### 4.1 EXTRACTING UNMEASURED CAUSAL FACTORS UNDER FORCING INPUTS

The Koopman Operator (KO) utilizes a measurement function $\mathcal{G} : \mathcal{R}^{|V|} \rightarrow \mathcal{R}^W$ to represent the data generation $\Psi$ as a linear combination of non-linear dynamics along $W$ dimensional manifolds: $\mathcal{KG}(V,U) = \mathcal{G}(\Psi(V,U))$. $\mathcal{K}$ is potentially infinite dimensional, i.e., $W \rightarrow \infty$. Most practical applications do not require infinite state space and are highly sparse in the manifold dimensions. If sparsity information is not available, then KO theory can be combined with MZ formalism to represent the non-linear dynamics as a set of Koopman observables $\mathcal{G}_M$ (data $X$ and $Y$) given by the measurement function and an orthogonal set of unmeasured causal factors $\mathcal{G}_I$ ($Z_c$ and $Z_d$) such that:

$$\begin{bmatrix} \dot{\mathcal{G}}_M \\ \dot{\mathcal{G}}_I \end{bmatrix} = \begin{bmatrix} \mathcal{K}_M & \mathcal{K}_{MI} \\ \mathcal{K}_{IM} & \mathcal{K}_I \end{bmatrix} \begin{bmatrix} \mathcal{G}_M \\ \mathcal{G}_I \end{bmatrix} , \mathcal{K}_M \text{ describes observable dynamics,} \mathcal{K}_I \text{ unmeasured causal factors, and } \mathcal{K}_{IM} \text{ interaction between the two.}$$

(4)

Utilizing Laplace transforms a solution for $\mathcal{G}_M$ can be obtained as follows

$$\dot{\mathcal{G}}_M = \underbrace{\mathcal{K}_M \mathcal{G}_M}_{\text{observable dynamics}} + \underbrace{\mathcal{K}_{MI} \int_0^t e^{t-s\mathcal{K}_I} \mathcal{K}_{MI} \mathcal{G}_M(s) ds}_{\text{interaction of causal factors with observable dynamics}} + \underbrace{\mathcal{K}_{MI} e^{-t\mathcal{K}_I} \mathcal{G}_I(0)}_{\text{Residual effect of causal factors}}$$

(5)

Sparse regression based techniques such as SINDY-MPC Kaheman et al. (2020) assumes all dynamics is explicitly measured and hence only works with observable dynamics.

**Steady state assumption:** Existing techniques Menier et al. (2025); Chen et al. (2021) for MR of spatio-temporal dynamics make an important assumption to simplify Eqn. 5. The process $P$ has been operating in steady state for time window of $[0, \tau]$. Under this assumption, Eqn. 5 becomes:

$$\dot{\mathcal{G}}_M = \mathcal{K}_M \mathcal{G}_M + \mathcal{K}_{MI} \int_{-\tau}^t e^{t-s\mathcal{K}_I} \mathcal{K}_{MI} \mathcal{G}_M(s) ds + \mathcal{K}_{MI} e^{-(t+\tau)\mathcal{K}_I} \mathcal{G}_I(-\tau)$$

(6)

If $\tau \rightarrow \infty$ then the last term of Eqn. 6 vanishes resulting in the following form:

$$\dot{\mathcal{G}}_M = \mathcal{K}_M \mathcal{G}_M + \mathcal{K}_{MI} \int_{-\infty}^t e^{t-s\mathcal{K}_I} \mathcal{K}_{MI} \mathcal{G}_M(s) ds$$

(7)

This steady state assumption thus enables existing techniques to learn the steady state effects of unmeasured causal factors through a high dimensional neural network such as recurrent neural networks Menier et al. (2025) as followed by existing techniques such as PINN+SR Chen et al. (2021), or Menier et al. (2025). These techniques thus have two components: a) unmeasured causal factors estimation in steady state through a deep network and Koopman state estimate through sparse regression such as SINDY Kaheman et al. (2020), which is then combined either in additive manner or through convolution.

**Inputs violate steady state assumptions:** One of the disadvantages of monitoring a process in steady state is that excitations of implicit factors can be nullified as the process reaches equilibrium states. Hence, existing PINN architectures are less likely to capture implicit causal factors dependent on exogenous variables.

We start from Eqn. 5, and instead of representing the nonlinear dynamics in terms of $t$ and $x$, the initial network

Figure 2: *GenEval* causal relation discovery method replicates MZ formulation using LTC-NN network layer.

layer represents the full Koopman state vector $[\dot{\mathcal{G}}_M, \dot{\mathcal{G}}_I]^T$ as a function of only observable states $\mathcal{G}_M$. An important property of this network is that it should mimic Eqn. 5 in presence of forcing inputs $u$. For simplicity we assume that the dynamics over the manifold is control affine, i.e. $\mathcal{G} = \mathcal{G}_f + \mathcal{G}_u u$, where $\mathcal{G}_f$ is the unperturbed dynamics while $\mathcal{G}_u$ is the input effects. We need a network that requires a forward pass of the following form:

$$\dot{\mathcal{G}}_M = \underbrace{\mathcal{K}_M (1 + \mathcal{G}_u u / \mathcal{G}_f + (\mathcal{G}_u + \mathcal{G}_I) u / \mathcal{G}_f) \mathcal{G}_f}_{\text{Input dependent time constant}} + \underbrace{\mathcal{K}_{MI} \int_0^t e^{t-s\mathcal{K}_I} \mathcal{K}_{MI} \mathcal{G}_f(s) ds + \mathcal{K}_{MI} e^{-t\mathcal{K}_I} \mathcal{G}_I(0)}_{\text{unmeasured causal factors}}$$

(8)

We achieve this by using a network of continuous time latent variable nodes (CTLV) in specific liquid time constant networks (LTC-NN) (Figure 2), whose forward pass is given by Eqn. 9.

$$\frac{dX}{dt} = -\frac{X}{\frac{\tau}{1+\tau G(V,U,\omega)}} + G(V,U,\omega),$$

(9)

here $G(.)$ is the output of the CTLV based representation layer comprised of LTC-NN nodes in Figure 2, $\omega$ is the weights of the CTLV layer, and $\tau$ is a time constant value of the LTC-NN node. The form of Eqn. 9 is the same as Eqn. 8 and **hence it becomes capable in learning the unmeasured causal factors under forcing inputs**. The estimated dynamics in the manifold space $\mathcal{G}$ is then passed into a dense nonlinear layer that models the inverse ODD solutions, where the function $\Psi$ is learned as the coefficients of the KO $\mathcal{K}_\Psi$. The output of the dense layer is passed through a sparsity guided dropout layer which reduces the number of terms in the estimated $\Psi$, i.e., reduce $|\mathcal{K}_\Psi|$, based on a threshold value much like SINDY-MPC Kaiser et al. (2018). The output of the dropout layer hence becomes the estimated $\Psi$, as a vector of coefficients $\mathcal{K}_\Psi$, the causal factor model of the data $X$. The estimated $\mathcal{K}_\Psi$ is then used to solve the ODD defined by the KO to obtain an estimation of $X$ using SOTA ODD solvers such as ODE45 Banerjee & Gupta (2024). The reconstruction error is used to train the entire network. We prove (in appendix) that architecture in Figure 2 is universal approximator of Eqn. 8.

### 4.2 Model free change detection in causal structures

We utilize the theory of conformal inference Tibshirani et al. (2019) to develop a distribution agnostic algorithm to identify whether an estimation of the model of causal factor $\Psi$ in terms of the coefficient of KO as $\mathcal{K}_\Psi(D_J)$, generated from a target domain $D_J$ is in the distribution of the set of models $\mathcal{K}_\Psi(D_I)$ learned from a source domain $D_I$. For any data point $X, Y \in D_J$, we define the robustness metric $\rho$ to quantify the difference of the causal factor model $\mathcal{K}_\Psi(X, Y)$ obtain from data $X, Y$ with the set of causal models obtained from data points in domain $D_I$ as in Eqn. 10.

$$\rho(\mathcal{K}_\Psi(X,Y), D_I) = \left(\sum_{i=1}^{|D_I|} \mathcal{K}_\Psi(X_i, Y_i)^T \mathcal{K}_\Psi(X,Y)\right)/|D_I|, \text{ where } X_i, Y_i \in D_I$$

(10)

where $|D_I|$ is the number of data points in the domain $D_I$ and $\mathcal{K}_\Psi(X,Y)^T$ denotes transpose of the KO operator coefficient vector. Given the robustness function $\rho(.,.)$ in Eqn. 10, conformal inference creates a prediction band $C \subset \mathcal{R}^2$ based on causal factor models from domain $D_I$ for a given *miscoverage level* $\alpha \in \{0, 1\}$, so that

$$Pr(\rho(\mathcal{K}_\Psi(X_i, Y_i), D_I) \in C) \geq 1 - \alpha, \text{ for any } X_i, Y_i \in D_I.$$

(11)

We call this interval $C$, domain conformal boundary (DCB) and is key to determine change in causal factor. If the robustness metric of the causal factor model $\mathcal{K}_\Psi(X, Y)$ from the data $X, Y \in D_J$ in target domain with respect to domain $D_I$ falls inside the interval $C$ for $D_I$, then $\mathcal{K}_\Psi(X, Y)$ does not have a causal factor that is not encompassed in $D_I$ with probability $(1 - \alpha)$. However, this is only for a single data point. Hence, we need an algorithm that extracts such an interval $C$ which extends the guarantee in Eqn. 11 to the entire target domain $D_J$.

Consider Algorithm 1 for DCB extraction, which takes the training data $(X_i, Y_i) \in D_I$ of source domain, miscoverage level $\alpha$ and the set of all causal factor model estimations $\mathcal{K}_\Psi(X_i, Y_i)$ as input and provides DCB as output.

---

**Algorithm 1 Domain Detect**($\{X_i Y_i\}_{i=1}^{|D_I|}$, $\alpha$, $\rho(.,.)$, $\mathcal{K}_\Psi(D_I)$)

1: **output** domain conformal boundary $C$
2: Split $D_I$ into two equal sized subsets $I_T$ and $I_V$.
3: Average robustness $\sigma = avg_i(\rho(\mathcal{K}_\Psi(X_i, Y_i), I_T/\{X_i, Y_i\}))$
4: For each $X_v, Y_v$ compute residual $R_j = \rho(\mathcal{K}_\Psi(X_v, Y_v), I_T) - \sigma$
5: **return** $d$ = the kth smallest value in $\{R_j : j \in I_V\}$, $k = \lceil(|I_V|/2 + 1)(1 - \alpha)\rceil$

---

The basic method is to divide the training set, data and their model estimates, into two mutually exclusive subsets $I_T$ and $I_V$. For each $\mathcal{K}_\Psi(X_i, Y_i) \in I_T$, $\rho(\mathcal{K}_\Psi(X_i, Y_i), I_T/\{X_i, Y_i\})$ is computed, where $I_T/\{X_i, Y_i\}$ denotes the set $I_T$ with $\{X_i, Y_i\}$ removed. Let $\sigma = avg_i(\rho(\mathcal{K}_\Psi(X_i, Y_i), I_T/\{X_i, Y_i\}))$ be the mean value of the robustness

---

**Algorithm 2 GenEval**($D_J$, $D_I$, $\rho(.,.)$, $\mathcal{K}_\Psi(D_J)$, $\mathcal{K}_\Psi(D_I)$, $\sigma$, $d$)

1: **input:** target domain $D_J$, source domain $D_I$, robustness $\rho$, causal model estimates, $\mathcal{K}_\Psi(D_J)$, $\mathcal{K}_\Psi(D_I)$, $\sigma$ and $d$ obtained from Algorithm 1.
2: **output** Percentage of $D_J$ with support violation
3: **for** each $\{X, Y\} \in D_J$ **do**
4:     Compute residual $R = \rho(\mathcal{K}_\Psi(X_i, Y_i), D_I) - \sigma$
5:     **if** $R_i \notin [-d, d]$ **then**
6:         add to the list of violations
7:     **end if**
8: **end for**
9: **return** percentage violation

---

metric in the training set $I_T$. From the validation set $I_V$ residual $\rho(\mathcal{K}_\Psi(X_v, Y_v), I_T) - \sigma$ is derived for every element in $I_V$, the residual is arranged in ascending order. The algorithm then finds the residual at the position $\lceil(|I_V|/2 + 1)(1 - \alpha)\rceil$ and is used as the prediction range $C$. We prove Theorem 1 that guarantees that the interval $C$ obtain from Algorithm 1 satisfies Eqn. 11.

**Theorem 1** *For any data point $X, Y \in D_J$, $Pr(\rho(\mathcal{K}_\Psi(X_i, Y_i), D_I) \in C = [-d, d]) \geq 1 - \alpha$, $\alpha > 0$ if and only if $Pr(\mathcal{K}_\Psi(X, Y)|\{X, Y\} \in D_J) = Pr(\mathcal{K}_\Psi(X, Y)|\{X, Y\} \in D_I)$, where $d$ is given by Algorithm 1. (Proof in appendix)*

### 4.3 Support violation detection algorithm

Utilizing Theorem 1 and Algorithm 1, we derived a robustness range that encodes the causal factors of source domain $D_I$. The *GenEval* mechanism in Algorithm 2 takes each sample from target domain $D_J$ and without labels and evaluates whether the sample is within the domain conformance boundary of $D_I$. It reports the percentage of datapoints in $D_I$ that violates support condition which is a measure of the degree of support violation of the target domain $D_J$ with respect to source domain $D_I$.

### 4.4 Invariance preservation violation detection

The method shown in Figure 1 can be applied to the representation space $g(X)|g : X \rightarrow S$ to determine invariance preservation. Given a learned hypothesis $h = f \circ g$, learned on the source domain $D_I$, Algorithm 1 and Algorithm 2 can be applied on $g(X)$ instead of $X$, $\forall \{X, Y\} \in D_J$ to determine the percentage of samples in $D_J$ for which $h$ violates the invariance preservation property based on Theorem 2.

**Theorem 2** *For any data point $X, Y \in D_J$, and hypothesis $h = f \circ g$ learned on data from source domain $D_I$, $Pr(\rho(\mathcal{K}_g(X, Y), D_I) \in C = [-d, d]) \geq 1 - \alpha$, $\alpha > 0$ if and only if $Pr(\mathcal{K}_g(X, Y)|\{X, Y\} \in D_J) = Pr(\mathcal{K}_g(X, Y)|\{X, Y\} \in D_I)$, where $d$ is given by Algorithm 1 with $X$ replaced by $g(X)$. (Proof in appendix)*

Here $\mathcal{K}_g(X, Y)$ is the causal factor model of $g(X)$ as obtained using the unmeasured causal factors extraction mechanism shown in Figure 2.

## 5 Evaluation and Results

We perform experiments with the following aims:

*Aim 1: evaluating unmeasured causal factor extraction accuracy* - **Benchmark:** We use benchmark time-series examples from Kaiser et al. (2018) for this purpose since these examples have known causal models (description in appendix). **Baselines:** comparators for *GenEval* in this evaluation are SINDY-MPC Kaiser et al. (2018) and PINN+SR Chen et al. (2021). PINN+SR specifically does not tackle forced dynamics. In this paper, we extended PINN+SR to tackle forcing inputs (details in appendix). **Metrics:** we use three metrics: root mean square error (RMSE), to determine data reconstruction accuracy from causal factor models, mean absolute relative difference (MARD), to determine accuracy of causal factor model coefficients, hamming distance (HD), to determine whether *GenEval* extracts the correct causal factors. **Experimental setup:** For each example, we took 100 samples with 10 iterations and report mean RMSE, MARD, and HD across the 10 trials.

*AIM 2: detecting causal support violation in simulation* - **Benchmark:** we chose two real world time series systems F8 cruiser aircraft system, and automated insulin delivery (AID) system and developed simulators based on Simulink for F8 and UVA/Padova type 1 diabetes (T1D) simulator Visentin et al. (2018). We generated two domains for each application with different causal factors. For F8 cruiser we created $F8Stuck$ dataset where the F8 simulator is simulated with its elevator stuck and $F8Normal$ with normal operation.

Table 2: RMSE, MARD. HD for time series benchmarks. Rows 1,2,3 all variables are measured, rows 4,5,6 all variables except $x_1$, the first state variable in each example are hidden.

| Method | Lorenz | Lotka | Pathogenic | F8 |
|---|---|---|---|---|
| SINDY-MPC | 0.2,4%,0 | 0.4,5%,1 | 1.3,13%,1 | 0.8,12%,3 |
| PINN+SR | 0.2,3%,0 | 0.3,3%,0 | 0.7,9%,0 | 0.8,10%,1 |
| *GenEval* | 0.2,2%,0 | 0.2,1%,0 | 0.7,9%,0 | 0.7,8%,1 |
| SINDY-MPC | 2.1,14%,3 | 4.3,25%,5 | 11.3,43%,7 | 9.3,34%,6 |
| PINN+SR | 2.3,13%,4 | 4.1,17%,4 | 10.9,37%,7 | 9.1,35%,6 |
| *GenEval* | 0.4,3.1%,11.2,4%,0 | | 5.1,11%,3 | 8.1,18%,3 |

For AID, we created $AIDRescue$ where an individual consumes 15 g of sugar whenever blood glucose goes below 70 mg/dl and $AIDNormal$ where no rescue carbs was consumed. A real world imaging application for diabetes retinopathy (DR) was used. We created two domains from APTOS, $DRNormal$ with no retinopathy, and $DRStage5$ with stage 5 retinopathy. Benchmark details in appendix. **Baselines:** We evaluate $> 10$ recent multivariate timeseries anomaly detection (MTAD) techniques available in Liu et al. (2024b). For DR we used the baselines implemented in Galappaththige et al. (2024). **Metrics:** Percentage of samples in $F8Stuck$ or $AIDRescue$ or $DRStage5$ that can be identified as a member of $F8Normal$ and $AIDNormal$, respectively also denoted as DCB%. **Experiments:** same as AIM 1.

*AIM 3: detecting previously unknown causal support violation in real world SDG / MDG scenario -* *Benchmark:* In time series modality, we chose Medtronic 670 G obtained from JAEB center JAEB center (2023) with two domains a) normal operation, and b) insulin cartridge failure mode. In image modality, we chose DR application with four domains from different centers. **Baselines:** For AID, we again compare with MTAD techniques in AIM 2. For DR, we compare with statistical data

distribution shift tests such as Kolmogorov-Smirnov (KS) test Berger & Zhou (2014). We compare DCB with baseline dataset difference metrics such as Maximum Mean Discrepancy (MMD) and Central Moment Discrepancy (CMD) Zellinger et al. (2017). **Metrics:** In AID, the metrics are same as AIM 2. In DR, with KS test we show the $p$ value that two domains significantly differ in distribution. For *GenEval*, we use percentage of data samples in one domain identified to be within DCB of other domain (DCB %). **Experiment setup:** For AID, we evaluate in SDG mode. For DR we evaluate in both SDG and MDG. For MDG, we create two datasets by combining three of the four datasets into a source domain and leaving one domain as target.

*AIM 4: detecting violation of NC1, invariant preserving representation condition* - **Benchmark:** We only test in the image based application of diabetes retinopathy (DR) using the four datasets of APTOS, EYEPACS, MESSIDOR 1 and MESSIDOR 2 Galappaththige et al. (2024). **Baselines:** The KS test. **Metrics:** same as AIM 3. **Experimental Setup:** We implemented visual transformers (ViT) Liu et al. (2023) and AlexNet CNN Yu et al. (2016) and evaluated their accuracy in MDG and SDG setup using the Domainbed framework Gulrajani & Lopez-Paz (2020). Simultaneously, *GenEval* was used on the representation layer of the ViT and AlexNet methods. For *GenEval*, we show the correlation of DCB % with the classification accuracy. In addition, utilizing the domainbed framework, we also test on SOTA five mainstream DG benchmarks and evaluate invariance preservation on 14 latest DG baselines discussed in Wen et al. (2025).

Table 3: Support violation in DR Single /Multi DG. Acc is max across all baselines. SPSD-ViT and SD-ViT are from(Galappaththige et al., 2024), DRGen(Atwany & Yaqub, 2022), and DANN from (Ganin et al., 2016)

| Source | Target | Acc.(%) | DCB (%) | Method | CMD | MMD |
|---|---|---|---|---|---|---|
| Messi dor-1 (M1) | A | $48.3 \pm 1.1$ | 25.15 | SPSD-ViT | 0.42 | 0.57 |
| | E | $57.4 \pm 2.1$ | 32.35 | SPSD-ViT | 0.47 | 0.49 |
| | M2 | $62.0 \pm 1.7$ | 34.25 | SD-ViT | 0.52 | 0.41 |
| Messi dor-2 (M2) | A | $52.8 \pm 2.0$ | 54.21 | SPSD-ViT | 0.45 | 0.6 |
| | E | $72.5 \pm 0.3$ | 63.41 | SPSD-ViT | 0.6 | 0.53 |
| | M1 | $46.7 \pm 0.1$ | 42.00 | SD-ViT | 0.39 | 0.55 |
| Aptos (A) | E | $72.0 \pm 0.8$ | 72.19 | SD-ViT | 0.59 | 0.47 |
| | M1 | $46.7 \pm 0.1$ | 58.58 | DRGen | 0.41 | 0.58 |
| | M2 | $61.0 \pm 0.1$ | 61.49 | DRGen | 0.5 | 0.51 |
| Eye pacs (E) | A | $75.1 \pm 0.5$ | 95.46 | SPSD-ViT | 0.63 | 0.63 |
| | M1 | $54.6 \pm 1.5$ | 82.96 | DRGen | 0.46 | 0.44 |
| | M2 | $65.4 \pm 0.1$ | 87.46 | DRGen | 0.55 | 0.59 |
| M1,M2,A | E | $73.6 \pm 0.3$ | 79.1 | SPSD-ViT | 0.62 | 0.46 |
| M1,M2,E | A | $54.4 \pm 0.8$ | 51.31 | DANN | 0.48 | 0.61 |
| M1,E,A | M2 | $73.3 \pm 0.2$ | 76.7 | SD-ViT | 0.61 | 0.5 |
| M2,E,A | M1 | $65.2 \pm 0.3$ | 75.6 | SPSD-ViT | 0.57 | 0.52 |

**AIM 1 results:** Table 2 shows that for the benchmark systems, *GenEval* performs on par with the baselines when all state variables are measurable. However, when only one variable is measurable then *GenEval* outperforms baselines on all metrics. This shows the capacity of modeling unmeasured causal factors.

**AIM 2 results:** There is a positive correlation of DCB % with accuracy of MTAD by the best baseline with pearson's correlation coefficient 0.91 p value 0.043. Comprehensive table is provided in appendix (Table 8).

Table 4: Support violation in AID SDG. Acc is max across all baselines

| Source | Target | Acc.(%) | DCB (%) | Baseline |
|---|---|---|---|---|
| Cartridge | Normal | 45.1 | 5.2 | TSB Liu & Paparrizos (2024) |
| Normal | Cartridge | 51.9 | 6.3 | MTADGAT Liu et al. (2024b) |

**AIM 3 results:** In Table 3, SDG experiments show that whenever DCB % is low then the maximum accuracy attained by any baseline is also low for that dataset. There is high correlation (pearsons correlation factor, 0.83, p value = 0.041) between DCB % and maximum accuracy. This shows that DCB can act as a surrogate in determining feasibility of DG on a target domain even without the need for access to the labels or even before application of any technique. If DCB is low for a target domain, then it implies that there is some new causal factor in target domain that is not available in source domains and hence any method trained in the source domain cannot perform well in the target domain. Similarly in the case of MDG, we observed that the domain conformance boundary for APTOS dataset is the largest ([-0.05, 0.05]), then Messidor 1 ([-0.045 0.045]), Messidor 2 ([-0.043 0.043]) and then EYEPACS ([-0.03 0.03]). Table 3 shows that whenever the source domain set does not have Aptos, the maximum attained accuracy for all baselines in Galappaththige et al. (2024) is lower than when APTOS is included in the source domain. This shows that the domain conformance boundary can be used to select the right set of source domains. Table 4 also shows similar results for the real world time series data in the AID cartridge failure application. The DCB % in the SDG setting is very low for the AID datasets. We also see that none of the existing MTAD techniques could achieve acceptable accuracy in detecting cartridge failure. Table 4 and 3 also shows in all SDG and MDG scenarios the KS test on the data distribution showed no change. We see that CMD and MMD are not correlated with Accuracy (0.503 and 0.211) unlike DCB.

Table 5: Invariance preservation violation in DR S/M DG.

| Source | Target | AlexNet Acc.(%) | AlexNet DCB (%) | ViT Acc | ViT DCB |
|---|---|---|---|---|---|
| Messi dor-1 (M1) | A | 21.8 | 15.15 | 43.2 | 41 |
| | E | 27.3 | 22.1 | 42.3 | 41 |
| | M2 | 62.0 | 37.6 | 64.1 | 56.2 |
| Messi dor-2 (M2) | A | 31.8 | 34.1 | 45.1 | 41.3 |
| | E | 42.9 | 51.2 | 51.5 | 46.1 |
| | M1 | 76.3 | 64.1 | 71.2 | 62.8 |
| Aptos (A) | E | 72.2 | 68.3 | 72.5 | 79.2 |
| | M1 | 77.6 | 71.8 | 78.1 | 81 |
| | M2 | 78.1 | 76.9 | 77.4 | 71 |
| Eye pacs (E) | A | 55.1 | 52.6 | 73 | 67 |
| | M1 | 45.5 | 50.1 | 64.1 | 61 |
| | M2 | 61.5 | 63.3 | 65.2 | 61 |
| M1,M2,A | E | 71.2 | 69.4 | 79.2 | 74.1 |
| M1,M2,E | A | 52.3 | 41.2 | 65 | 54 |
| M1,E,A | M2 | 67.1 | 66.6 | 71.2 | 69.1 |
| M2,E,A | M1 | 62.5 | 64.1 | 64.1 | 66.2 |

**AIM 4 results:** From Table 5 we see that for a hypothesis (AlexNet or ViT) if DCB % is low the accuracy for either SDG or MDG is also low indicating high correlation with accuracy and DCB %. A low DCB % results in violation of invariance preservation property. Hence, application of *GenEval* results in the DCB % metric that is effective in determining invariance preser-

Table 6: Execution Time Comparison on NVIDIA GeForce RTX 3080 (Eyepacs Dataset, 33K Samples, Extrapolated to 100K Images)

| Method | Wallclock time (s) | Time per Sample (s) | Extrapolated to 100K Images |
|---|---|---|---|
| AlexNet | 8 hrs 23 mins $\pm$ 32 mins | 0.91 | $\approx$ 25 hrs 24 mins |
| ViT | 16 hrs 29 mins $\pm$ 43 mins | 1.80 | $\approx$ 49 hrs 57 mins |
| DCB | 37 mins $\pm$ 7 mins | 0.07 | $\approx$ 1 hr 52 mins |

vation violation. Table 10 in Appendix shows the DCB for each SOTA baseline on five large scale benchmark datasets. We see here that DCB has good correlation (0.869 p value of 0.056) with the maximum achievable accuracy in each benchmark. This shows that the GenEval framework is generic and can guide model selection and source domain selection on large scale DG datasets.

**Computational Complexity and Execution time:** There are two parts to GenEval: a) Finding the $[-d, d]$ range of the source: This requires identifying the causal factors and then running Algorithm 1 (Domain Detect). b) Finding the DCB between source and target: This is performed using Algorithm 2 (GenEval). In part (a), for the LTC-NN, the computational complexity of the forward pass is $O(V + V(|\Theta| + q)) + O(|X|N)$, where $V$, $q$, $\Theta$, and $X$ are as defined in Fig.2. The complexity of the backward pass is $O(V P_{\text{LTC}} N + V(|\Theta| + q) P_{\text{dense}} N)$, where $P_{\text{LTC}}$ is the number of parameters in the LTC cell and $P_{\text{dense}}$ is the number of parameters in each neuron of the dense layer. Each operation involves a multiplication. The complexity of Algorithm 2 is $O(|D_I|) + O(|D_J|)$, where $|D_I|$ and $|D_J|$ denote the number of samples in the source and target domains respectively. Each operation corresponds to computing a cosine similarity metric. DCB exhibits significant speedup compared to inference times of AlexNet ($12\times$) and ViT ($25\times$) on large-scale datasets (Table 6).

**Testing the differentiability assumption:** Continuous differentiability is required only for the causal factor recovery stage. Discontinuities (e.g., edges or hard boundaries) can be handled via boundary conditions and reformulated as compartmental state–space models, where Koopman-based analysis and our framework remain applicable, although excessive discontinuities may

Table 7: DCB on Categorical Tabular Datasets

| Task | Target | Shift Domain | Acc diff % | DCB |
|---|---|---|---|---|
| ASSISTments | Next Answer Correct | School | $-34.49$ | 68.64 |
| Scorecard | Degree Completion Rate | Institution Type | $-11.16$ | 57.76 |
| ICU Hospital | ICU Patient Expires | Insurance Type | $-6.30$ | 48.87 |
| Readmission | 30-Day Readmission | Admission Source | $-5.94$ | 51.54 |
| Diabetes | Diabetes Diagnosis | Race | $-4.48$ | 55.67 |
| ICU Length | ICU Stay $>$ 3 hrs | Insurance Type | $-3.39$ | 49.87 |
| Voting | Election Voting | Geographic Region | $-2.58$ | 54.10 |
| Food Stamps | Food Stamp Recipiency | Geographic Region | $-2.39$ | 51.00 |
| Unemployment | Unemployment Status | Education Level | $-1.28$ | 53.77 |
| Income | Income $\geq$ \$56K | Geographic Region | $-1.25$ | 38.60 |

cause state–space explosion. Importantly, our core metric, **DCB**, is independent of continuity assumptions and operates solely on recovered causal factors, making it agnostic to the extraction mechanism. Thus, DCB remains valid even for discrete or non-differentiable data-generating processes given an appropriate recovery method. We demonstrate this on categorical tabular datasets from TableShift Gardner et al. (2023), where each column is treated as a causal factor. Using the domain-shift accuracy changes reported in Table 7, we compute DCB for each task. We observe a strong negative correlation between DCB and accuracy change ($-0.8$, , $p = 0.02$), indicating that higher DCB values correspond to smaller performance degradation under domain shift, supporting its validity as a robustness indicator even in non-differentiable settings.

**Category shift evaluation:** In this experiment, we evaluate whether GenEval can identify category shift, which is a more severe form of causal cover violation Fu et al. (2020). In the DR example, we simulate category shift by randomly removing one of five classes from each source domain and evaluating in target domain on all 5 classes. We compare the performance of DCB in detecting category shift with H-score proposed in (Fu et al., 2020). Table 9 in Appendix shows that the correlation between DCB and H-score (0.924) is higher than the correlation between DCB and Accuracy (0.91). This reaffirms the applicability of DCB as a category shift identifier.

## 6 CONCLUSIONS

The task of domain generalization (DG) is an important task for the practical deployment of any AI technique. As such it is the Achilles Heel of machine learning especially in the medical imaging domain. In this regard, *GenEval* answers two important questions on the feasibility of generalization into a target domain: a) does the set of training source domains provide causal support? and b) does the learned machine accurately characterize all causal factors? *GenEval* can also help researchers to take informed decision on the selection of source domains. This can pave the way for a theoretical analysis of DG problem and novel solutions grounded by necessary and sufficient conditions. **Limitations:** *GenEval* assumes that the data generation process is continuous and differential. While this may not be true in real world especially with interaction of computing and physical systems, any non-differentiable system can be approximated as a switched hybrid differentiable systems around the discontinuities. However, further research is needed to address this. The DCB metric may possibly be manipulated through adversarial augmentation of source or target data. A nuanced analysis is needed to verify under what circumstances DCB can be manipulated which is an interesting extension.

## 7 REPRODUCIBILITY

All code is shared via supplementary document.

## 8 ETHICS STATEMENT

GenEval is an evaluation framework for assessing domain-generalization feasibility (e.g., support and representation-invariance) without target labels; it is not a diagnostic or operational system, and any safety-critical use requires IRB/ethics review, domain oversight, and prospective validation. We use only publicly available or institutionally released, de-identified datasets under their licenses; no new human data were collected and no re-identification was attempted. To mitigate bias, we report per-domain results, document dataset composition and preprocessing, and encourage diversity in sources/devices so subpopulations are not disadvantaged. Privacy and security are respected by excluding protected health information from released artifacts and cautioning against membership/attribute inference; GenEval's feasibility flags must not be used to justify skipping target-domain data collection—negative flags warrant more data and validation, while positive signals are not guarantees. We will release code and configs for reproducibility and favor moderate-footprint computing (e.g., mixed precision, checkpoint reuse) to reduce environmental impact.

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

## A  DESCRIPTION OF BENCHMARKS IN AIM 1

The benchmarks examples used in AIM 1 of the paper are shown in this section along with the models extracted by *GenEval*. All coefficients are rounded up to three decimal points after most significant digit.

**Lotka Volterra** : It has two variables $x_1$ and $x_2$ given by the following equations:

$$\dot{x_1} = ax_1 - bx_1x_2, \ \dot{x_2} = -cx_2 + dx_1x_2 + u$$

a = 0.5, b = 0.025, c = 0.5, and d = 0.005

**Recovered model with all variables measured:**

$$\dot{x_1} = 0.52x_1 - 0.026x_1x_2, \ \dot{x_2} = -0.501x_2 + 0.005x_1x_2 + 0.999u$$

**Chaotic Lorenz**: The chaotic lorenz system is described in the following equations:

$$\dot{x_1} = \sigma(x_2 - x_1) + u, \dot{x_2} = x_1(\rho - x_3) - x_2, \dot{x_3} = x_1x_2 - \beta x_3,$$

$\sigma = 10$, $\beta = 8/3$, $\rho = 28$.

**Recovered model with all variables measured:**

$$\dot{x_1} = 10.000(x_2 - x_1) + 0.999u, \ \dot{x_2} = 27.992x_1 \check{}\ 1.002x_1x_3 \check{}\ 0.998x_2, \ \dot{x_3} = 1.000x_1x_2 \check{}\ 2.7x_3$$

**F8 Cruiser**: The F8 Cruiser system is given by:

$$\dot{x_1} = -0.9x_1 + x_3 - 0.09x_1x_3 + 0.47x_1^2 - 0.02x_2^2 - x_1^2x_3 + 3.85x_1^3 - 0.21u + 0.28x_1^2u + 0.47x_1u^2 + 0.6u^3$$
$$\dot{x_2} = x_3, \dot{x_3} = -4.208x_1 - 0.396x_3 - 0.47x_1^2 - 3.564x_1^3 - 20.967u + 6.265x_1^2u + 46x_1u^2 + 61.1u^3$$

**Recovered model with all variables measured:**

$$\dot{x_1} = -0.872x_1 + 0.998x_3 - 0.088x_1x_3 + 0.476x_1{}^2 - 0.0186x_2{}^2 \check{}\ 0.970x_1{}^2x_3$$
$$+ 3.849x_1{}^3 - 0.22u + 0.265x_1{}^2u + 0.472x_1u^2 + 0.63u^3, \ \dot{x_2} = 1.000x_3$$
$$\dot{x_3} = -4.210x_1 - 0.399x_3 - 0.465x_1{}^2 - 3.565x_1{}^3 - 20.978u + 6.267x_1{}^2u + 45.711x_1u^2 + 62.002u^3$$

**Pathogenics attack model**: The pathogenic attack system is given by:

$$\dot{x_1} = \lambda - dx_1 - \beta(1 - \eta u)x_1x_2, \ \dot{x_2} = \beta(1 - \eta u)x_1x_2 - ax_2 - p_1x_4x_2 - p_2x_5x_2$$
$$\dot{x_3} = c_2x_1x_2x_3 - c_2qx_2x_3 - b_2x_3, \ \dot{x_4} = c_1x_2x_4 - b_1x_4, \ \dot{x_5} = c_2qx_2x_3 - hx_5,$$

with $\lambda = 1$, $d = 0.1$, $\beta = 1$, $a = 0.2$, $p_1 = 1$, $p_2 = 1$, $c_1 = 0.03$, $c_2 = 0.06$, $b_1 = 0.1$, $b_2 = 0.01$, $q = 0.5$, $h = 0.1$, and $\eta = 0.9799$.

**Recovered model with all variables measured:**

$$\dot{x_1} = 0.939 - 0.1x_1 - 0.982x_1x_2 + 0.98ux_1x_2, \ \dot{x_2} = 0.982x_1x_2 - 0.98ux_1x_2 - 0.18x_2 - 1x_4x_2 - 1.001x_5x_2$$
$$\dot{x_3} = 0.059x_1x_2x_3 - 0.03x_2x_3 - 0.009x_3, \ \dot{x_4} = 0.029x_2x_4 - 0.1x_4, \ \dot{x_5} = 0.059x_2x_3 - 0.1x_5$$

## B  ARCHITECTURE IN FIGURE 2 IS UNIVERSAL DYNAMICS APPROXIMATOR

The forward pass of liquid time constant neural network (LTC-NN) is given by Hasani et al. (2021):

$$\frac{dg(t)}{dt} = -g(t)/\tau + f_{NN}(g(t), I(t), t, \omega)(A - g(t)), \tag{12}$$

where $g(t)$ is one hidden state of the LTC-NN, $\tau$ is a time constant parameter, required to assist any autonomous system to reach equilibrium state. As such existence of the $-g(t)/\tau$ is an important stability criteria as it ensures that the unperturbed plant settles in time. $f_{NN}$ is the forward pass and is a function of the hidden states, $I(t)$ is the input to the LTC-NN, $\omega$ and $A$ are the parameters of the LTC-NN architecture.

**Theorem 3** *The forward pass of an LTC-NN architecture generates a set of implicit physical dynamics that are equivalent to a bilinear approximations of the control affine autonomous system in Eqn. 8.*

**Proof:** Algebraic manipulation of the forward pass of LTC-NN architecture gives the structure of Eqn. 13 which allows an input dependent time constant $\frac{\tau}{1+\tau f_{NN}(g(t),I(t),t,\omega)}$.

$$\frac{dg(t)}{dt} = -\frac{g(t)}{\frac{\tau}{1+\tau f_{NN}(g(t),I(t),t,\omega)}} + f_{NN}(g(t), I(t), t, \omega)(A). \tag{13}$$

The stability criteria for any autonomous system requires the control affine model to have a time constant term as shown in Eqn. 14

$$\frac{dX}{dt} = -X/\tau + \mathcal{G}_{f_{-\tau}}(X) + \mathcal{G}_u(X)U_T, \tag{14}$$

where $\tau$ is the time constant of the system and $\mathcal{G}_{f_{-\tau}}(.)$ is the unperturbed dynamics obtained by removing the time constant component from $\mathcal{G}_f(.)$.

Assuming that the autonomous system is a dynamic causal system, the bilinear approximation Friston et al. (2003) of the control affine system in Eqn. 14 results in Eqn. 15.

$$\frac{dX}{dt} \approx -X/\tau + \mathcal{G}_{f_{-\tau}}(X) + BX + CU_T + \sum_j u_T^j D^j X + H, \tag{15}$$

where $B = \frac{\partial(\mathcal{G}_u(X)U_T)}{\partial X}$, $C = \frac{\partial(\mathcal{G}_u(X)U_T)}{\partial U_T}$, and $D^j = \frac{\partial^2(\mathcal{G}_u(X)U_T)}{\partial X \partial u_T^j}$, $H$ is a constant. Rearranging Eqn. 15, we have the similar form as the LTC-NN forward pass in Eqn. 16.

$$\frac{dX}{dt} \approx -\frac{X}{\frac{\tau}{1+\tau(B+\sum_j u_T^j D^j)}} + (\mathcal{G}_{f_{-\tau}}(X) + CU_T + H). \tag{16}$$

We observe that Eqn. 16 is the same form as Eqn. 13 if the input to the LTC-NN $I(t)$ is a concatenation of $Y$ and $U_T$. The hidden layers of the LTC-NN model an inflated set of implicit dynamics which may include the unmeasured system variables of the physics model.

**Theorem 4** *The inflated set of implicit dynamics modeled by LTC-NN induces an over-determined set of equations in the coefficients of the bilinear approximation of any control affine model.*

**Proof:** The training process of LTC-NN fixes weights and instantiates the hidden layer outputs. The values of the unmeasured variables in $X$ is estimated by the hidden state in each training step utilizing the forward pass and learned LTC-NN weights $\omega$. Hence each forward pass provides an over-determined set of linear equations in the coefficients $B$, $C$, and $D^j$.

The original control affine model coefficients $\Theta$ are non-linear functions of the coefficients $B$, $C$, and $D^j$s, The dense layer is best suited for exploring a large set of possible non-linear combinations of $B$, $C$, and $D^j$ that express $\Theta$. An overdetermined system of equations is inconsistent and may be unsolvable. The dense layer guided by the ODE solver induced loss function (ODE Loss) learns a consistent set of linear equations in $B$, $C$, and $D^j$ and also learns their non-linear combination to determine $\Theta$.

## C    PROOF OF THEOREM 1

For any data point $X, Y \in D_J$, $Pr(\rho(\mathcal{K}_\Psi(X_i, Y_i), D_I) \in C = [-d, d]) \geq 1 - \alpha, \alpha > 0$ if and only if $Pr(\mathcal{K}_\Psi(X, Y)|\{X, Y\} \in D_J) = Pr(\mathcal{K}_\Psi(X, Y)|\{X, Y\} \in D_I)$, where $d$ is given by Algorithm 1.

**Proof:** To prove the if part of Theorem 1 lets assume that $Pr(\rho(\mathcal{K}_\Psi(X_i, Y_i), D_I) \in C = [-d, d]) \geq 1 - \alpha, \alpha = 0.05$. This means that $[-d, d]$ is the 95% confidence interval of the robustness value $\rho(\mathcal{K}_\Psi(X_i, Y_i), D_I)$. Now $[-d, d]$ is derived from the training data of $D_I$. Which means that $[-d, d]$ is also the 95% confidence interval of $\rho(\mathcal{K}_\Psi(X, Y), D_I)$ for $\forall X, Y \in D_I$. This entails that the 95% confidence intervals of $D_I$ and $D_J$ are the same. The interval is a zero mean distribution, hence any memory-less stochastic process can uniquely define the interval with only one parameter, $\alpha$. Since $\alpha$ only defines the probability density and $\alpha$ is same for both $X_i, Y_i \in D_J$ and $X, Y \in D_I$ we prove that $Pr(\mathcal{K}_\Psi(X, Y)|\{X, Y\} \in D_J) = Pr(\mathcal{K}_\Psi(X, Y)|\{X, Y\} \in D_I)$.

Note that the interval was defined by $\alpha$ where $\alpha$ can be chosen arbitrarily. This implies that if $\forall \alpha$ the intervals are same, then even for non memoryless distributions, the two distribution have to be same.

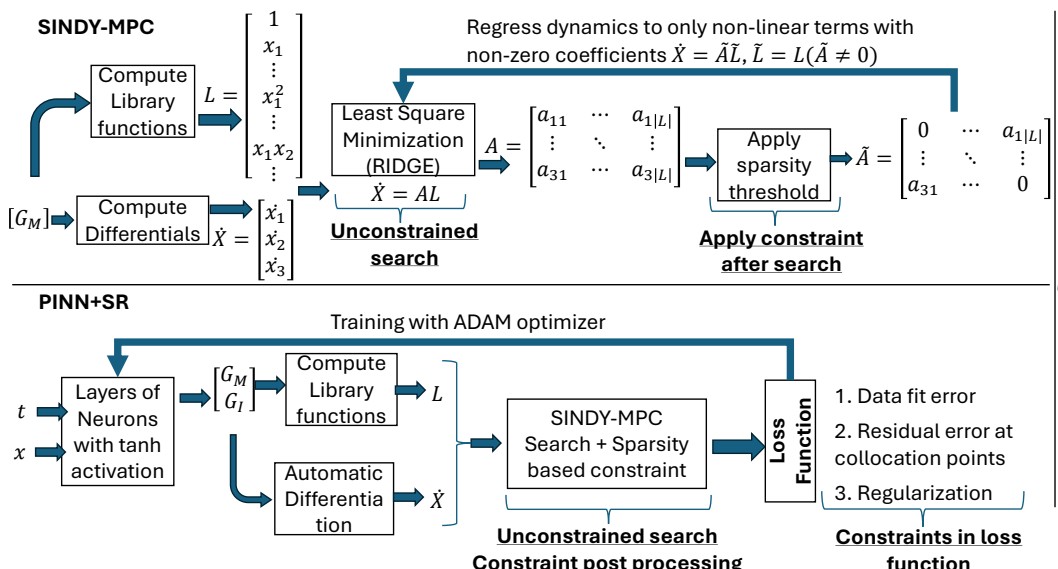

Figure 3: Learning architectures of SINDy-MPC and PINN+SR.

The only if part is simpler since if the distributions are same, the $1 - \alpha$ confidence intervals will be same by definition.

## D  PROOF OF THEOREM 2

For any data point $X, Y \in D_J$, and hypothesis $h = f \circ g$ learned on data from source domain $D_I$, $Pr(\rho(\mathcal{K}_g(X,Y), D_I) \in C = [-d, d]) \geq 1 - \alpha, \alpha > 0$ if and only if $Pr(\mathcal{K}_g(X,Y)|\{X,Y\} \in D_J) = Pr(\mathcal{K}_g(X,Y)|\{X,Y\} \in D_I)$, where $d$ is given by Algorithm 1 with $X$ replaced by $g(X)$.

**Proof:** This theorem is only arrived at if the two domains $D_I$ and $D_J$ have the same set of causal factors, i.e. Theorem 1 is satisfied. In that case the proof of this theorem is straightforward.

If $X, Y \in D_I$ and $X_i, Y_i \in D_J$ are random variables with $Pr(\mathcal{K}_\Psi(X,Y)|\{X,Y\} \in D_J) = Pr(\mathcal{K}_\Psi(X,Y)|\{X,Y\} \in D_I)$, then for any measurable function $g \colon \mathbb{R}^{|X|} \to \mathbb{R}^M$, where $M$ is the dimension of the representation $g(X,Y)$ and $g(X_i,Y_i)$ have the same distribution. $\mathcal{K}_g(.)$ is the potentially infinite dimensional koopman operator and hence can represent the function $g(.)$ with arbitrary precision. Hence properties on $g(.)$ will also satisfy on $\mathcal{K}_g$. Hence, as a consequence of measurability and satisfaction of Theorem 1, Theorem 2 is proved.

## E  IMPLEMENTATION DETAILS OF SINDY-MPC AND PINN+SR

**SINDy-MPC:** Significant breakthrough was achieved through introduction of sparse identification of non-linear dynamics (SINDy). Subsequently SINDy has been extended to tackle control inputs in SINDy-MPC Kaiser et al. (2018), however, as shown in this manuscript, it does not generalize well for low sampling frequencies. Given $N$ samples of data, at sampling frequency $f_r = \frac{1}{\tau}$ the recovery problem can be reduced to solving the following set of linear equations:

$$\begin{bmatrix} \frac{dX}{dt}(\tau) \\ \vdots \\ \frac{dX}{dt}(N\tau) \end{bmatrix} = \begin{bmatrix} \zeta(X(\tau),0) & \cdots & \zeta(X(\tau),W) \\ \vdots & \vdots & \vdots \\ \zeta(X(N\tau),0) & \cdots & \zeta(X(N\tau),W) \end{bmatrix} \times \begin{bmatrix} \theta_0 \\ \vdots \\ \theta_W \end{bmatrix}, \tag{17}$$

where $W = \binom{M+n}{n}$, and $\zeta(X(k\tau), i)$ is the $i^{th}$ term in the bilinear expansion of $f(X)$ at the time sample $t = k\tau$. For sparse identification, majority of the $\theta_j \approx 0, j \in \{1 \ldots W\}$ with only $p$ significant elements $\Theta = [\theta_1 \ldots \theta_p]$. As such $N >> p$, making Eqn 17 an over-determined set of linear equations with no consistent solution. A solution method is least squares minimization, to recover $\Theta_{est}$ that minimizes $e_T = ||X_{est} - X||^2$. SINDy-MPC solves the unconstrained least square minimization problem using the sequential threshold ridge regression (STRidge) algorithm (Figure

3). This method iteratively selects dominant candidate from a library of high dimensional nonlinear functions Quade et al. (2018). The sparsity was achieved through iteratively removing non-linear components utilizing hard thresholds on the derived model coefficients.

**PINN+SR:** As shown in Figure 3, the PINN+SR approach starts with generating a valuation of the multi-variate data $X$ as a function of time using a feedforward network Chen et al. (2021). The derivatives of $X$ with respect to time are then derived by another layer of nodes with automatic differentiation. The derivatives and valuations of $X$ are used to evaluate a library of nonlinear functions. These non-linear functions are provided as input to the SINDy method that outputs a sparse matrix using the STRIdge method. The sparse matrix is used to re-evaluate the solution of the ODE and compute residual with respect to the ground truth, the physics loss, and regularization loss. The overall loss is used to optimize and train the weights of the entire network using backpropagation.

## F    AIM 2 BENCHMARK DETAILS

**F8Cruiser:** This is an aircraft pitch control system using a model predictive control for trajectory tracking. The first domain $D_1$ is normal pitch control operation. The second domain $D_2$ is a **hardware failure** where the elevator gets jammed and maintains a constant position overriding the controller ($F8Stuck$). We created another domain $F8Slow$ where the elevator speed is reduced by 3 times the normal speed.

**Automated Insulin Delivery System** This is an hybrid close loop autonomous system that autonomously decides on insulin delivery for the most part, but requires **human intervention** with extra insulin delivery to manage meal intake. The human may trick the system to deliver a high dosage of insulin by announcing to the system that a large meal has been ingested without actually consuming the meal. Domain $D_1$ is normal operation of the AID. The domain $D_2$ has data from individuals who injested rescue meal of carbohydrate content of 15 g whenever glucose is less than 15 g ($AIDRescue$). The underlying causal factor model of the AID system is govered by the bergman minimal model (BMM) described in Eqn. 18. We created another domain $D_3$ with insulin cartridge failure $AIDCartridge$. In cartridge failure, whenever the insulin system needs to inject bolus, it does not inject. Instead it builds up at the cartridge. Finally, when the pressure goes beyond a threshold, a large dosage of insulin rushes through into the blood stream.

$$\dot{\delta i}(t) = -n\delta i(t) + p_4 u_1(t) \tag{18}$$

$$\dot{\delta i_s}(t) = -p_1 \delta i_s(t) + p_2(\delta i(t) - i_b) \tag{19}$$

$$\dot{\delta G}(t) = -\delta i_s(t)G_b - p3(\delta G(t)) + u2(t)/VoI, \tag{20}$$

The input vector $U(t)$ consists of the input insulin level $u_1(t)$ and the glucose appearance rate in the body $u_2$. The output vector $Y(t)$ comprises the blood insulin level $i$, the interstitial insulin level $i_s$, and the BG level $G$. In AP, only $G$ is a measurable output. $i_s$ and $i$ are un-measurable. $p_1, p_2, i_b, p_3, p_4, n,$ and $1/V_oI$ are all patient specific coefficients.

**Unmanned Aerial Vehicle:** This is a quadcoptor, whose altitude is controlled by four proportional integrative and derivative (PID) controllers. These controllers provide balanced thrusts in each propeller so that the UAV maintains a given height. The first domain $D_1$ is normal altitude control operation of the UAV. The second domain $D_2$ is a **software failure** that changes the gravity parameter $g$ in the controller software ($UAVSimG$). The third domain $D_3$ is an **electromagnetic attack** on the UAV gyroscope sensor ($UAVEMA$).

**Diabetic Retinopathy:** It is an important image classification application in the medial domain. There have been several works in DR that attempts to classify state 0 i.e. no DR to stage 5 i.e. proliferative DR. We create two domains $DRNormal$ for stage 0 DR and $DRStage5$ with stage 5 DR.

## G    AIM 2 COMPREHENSIVE RESULTS

For the time series case studies, in each of the causal support violation case study, we train the baseline technique as an autoencoder on 80% of the normal data. Then we extract the intermediate representation from the 20% normal data and domain $D_2$ or $D_3$ data. We then extract an anomaly score from the internal representation and use extreme value theory to determine if the datapoint is normal or from a different domain. We then report the precision recall and F1 score for each technique in Table 8.

For the image domain datasets, the baseline techniques were already configured to classify five classes of DR in Galappaththige et al. (2024). We re-ran the baseline techniques to determine the precision recall and accuracy for Stage 5 DR and Stage 0 DR classification only. This is reported in

Table 8. We then extract the DCB% from each domain for each case study with respect to the normal data $D_1$ for the given case study. This is reported in the DCB row of Table 8. As seen the maximum F1 score obtained in Table 8 is highly correlated with the DCB % (correlation coefficient 0.91 p = 0.043). This shows that the DCB is a good indicator of the necessary conditions for DG, that requires that a domain should not introduce new causal factor. If it does introduce new causal factor then DCB % decreases as a consequence, accuracy (F1 in this case) of machines also decrease.

## H  CATEGORY SHIFT TABLE

Here is the category shift Table discussed in results.

## I  DOMAINBED COMPREHENSIVE RESULTS

## J  CODE ORGANIZATION WITH SINDY-MPC AND DR EXAMPLE

### J.1  THETA DATA GENERATION AND PREPROCESSING

In this study, we utilized four well-established ophthalmic imaging datasets—Messidor-1 (M1), Messidor-2 (M2), Aptos 2019 (Aptos), and EyePACS—which are prominently used in diabetic retinopathy (DR) research. To extract meaningful latent representations, we implemented an advanced preprocessing and data-generation pipeline leveraging the PySINDy framework, designed for sparse identification of nonlinear dynamical systems.

Initially, each retinal fundus image was standardized by resizing to a uniform resolution of $512 \times 512$ pixels, followed by applying a circular mask to isolate the fundus region and reduce irrelevant peripheral noise. Image contrast was further enhanced using Contrast Limited Adaptive Histogram Equalization (CLAHE) within the LAB color space, significantly improving the visibility of subtle retinal features critical for downstream analysis.

Next, a unique radial sampling strategy was executed, wherein each preprocessed image was sampled in concentric circles starting from the central region outward, with sampling points placed at consistent angular intervals (every $10°$). This method generated a structured radial time series of RGB values, capturing how the retinal characteristics evolve from the image center toward the periphery.

The resulting radial trajectories served as input time-series data for the PySINDy model, enabling the discovery of underlying dynamical relationships inherent within retinal structures. For each sampled trajectory, the following computational steps were performed:

1. Construct a time-series matrix $X$ from the radial RGB trajectories.
2. Compute the time derivative $dX$ using forward finite differences.
3. Apply the PySINDy framework using a third-degree polynomial feature library and a sparse thresholded least squares optimizer.
4. Extract sparse model coefficients (referred to as $\theta$-values), representing latent dynamics describing radial RGB transitions.

Robustness and scalability were integral to our implementation, demonstrated by processing images in batches of 1000, incorporating intermediate result checkpointing, systematic memory management, detailed error logging, and the ability to resume processing seamlessly after interruptions. The generated $\theta$-values, which succinctly encapsulate significant retinal features and underlying image dynamics, were stored alongside metadata—image identifiers, error metrics, and model performance measures—in organized output directories, providing a solid foundation for subsequent analyses.

### J.2  DOMAIN CONFORMAL BOUNDARY COMPUTATION

After collecting the $\theta$-vectors from PySINDy, we applied a specialized "RHO" analysis to each dataset (M1, M2, Aptos, EyePACS) to compute a conformal boundary quantifying cross-domain alignment. The procedure was:

1. **Data loading and split.** Load all $\theta$-vectors (size $N \times 128$) and split into training ($Q_{\text{train}}$, 60%) and validation ($Q_{\text{val}}$, 40%) sets.
2. **Leave-one-out $\rho$-value (training).** For each $\theta_k \in Q_{\text{train}}$:
   (a) Define $Q_{\text{train}}^{-k} = Q_{\text{train}} \setminus \{\theta_k\}$.

Table 8: Comparison of **SPIE-AD** against baseline techniques for U2 benchmark examples (R is real world, S is synthetic). **SPIE-ADS** uses SINDY-MPC, while **SPIE-ADL** uses LTC-NN. [+] denotes with point adjustment (PA) and absence of [+] is without PA.

| Approach | F8Stuck S | | | F8Slow S | | | UAVSimG S | | | UAVEMA S | | | AIDPhantom S | | | AIDCartridge S | | | DRStage5 | | |
|---|---|---|---|---|---|---|---|---|---|---|---|---|---|---|---|---|---|---|---|---|---|
| | Pr | Re | F1 | Pr | Re | F1 | Pr | Re | F1 | Pr | Re | F1 | Pr | Re | F1 | Pr | Re | F1 | Pr | Re | F1 |
| Omni Su et al. (2019) | 41 | 26.8 | 32.4 | 65 | 28.1 | 39.2 | 32 | 19.7 | 24.4 | 29 | 16.8 | 21.3 | 19.1 | 16.5 | 17.7 | 65 | 31.9 | 43 | | | |
| AT Xu et al. (2022) | 85.5 | 75.8 | 80.3 | 34.2 | 32.8 | 33.5 | 35 | 33.5 | 34.2 | 33.9 | 32.4 | 33 | 34 | 32 | 33 | 34.3 | 33.8 | 34 | | | |
| iForest Liu et al. (2008) | 14 | 33 | 19.6 | 9.8 | 8.2 | 8.9 | 10.6 | 8.5 | 9.4 | 8.6 | 7.6 | 8.1 | 9.5 | 8.1 | 8.7 | 9.5 | 7.9 | 8.6 | | | |
| LODA Pevný (2016) | 88 | 70 | 78 | 60.7 | 13.7 | 22.4 | 50.7 | 11 | 18 | 35 | 8.6 | 13.8 | 35.8 | 9.4 | 14.9 | 36.4 | 9.7 | 15.3 | | | |
| LSTM Hundman et al. (2018) | 77 | 85 | 80 | 61 | 35.8 | 45.2 | 59.4 | 13.2 | 21.6 | 60.8 | 14.2 | 23 | 58.6 | 12.6 | 20.7 | 54.7 | 12.1 | 19.9 | | | |
| USAD Audibert et al. (2020) | 81 | 67.7 | 74 | 55.3 | 14.2 | 22.6 | 51.2 | 12.3 | 19.8 | 49.2 | 12.1 | 19.4 | 52.6 | 12.1 | 19.7 | 58 | 8.8 | 15.2 | | | |
| GANF Zhao et al. (2022) | 61 | 79 | 68.8 | 3.2 | 4.3 | 3.7 | 51.4 | 85 | 64.3 | 0.9 | 24.7 | 1.8 | 3.2 | 4.5 | 3.8 | 2.1 | 2.7 | 2.4 | | | |
| GAT Zhou et al. (2020b) | 71.4 | 80.5 | 75.7 | 58.9 | 34.5 | 43.5 | 59.2 | 32.3 | 41.8 | 50.4 | 28 | 36 | 54.5 | 28.9 | 37.8 | 57.2 | 30.3 | 39.7 | | | |
| OFA Zhou et al. (2023) | 21.4 | 4.5 | 7.4 | 21.9 | 9.7 | 13.4 | 37.5 | 22.1 | 27.2 | 20.3 | 8.5 | 12 | 31.3 | 18.3 | 23.1 | 21.7 | 10.1 | 13.8 | | | |
| FITS Xu et al. (2024) | 21.4 | 8.6 | 12.3 | 48.1 | 14.3 | 22.05 | 17.3 | 21.9 | 19.3 | 80.4 | 2.4 | 4.7 | 24.5 | 18.4 | 21.0 | 14.7 | 40.1 | 21.5 | | | |
| TFAD Zhang et al. (2022) | 11.2 | 30.4 | 16.4 | 9.8 | 21.7 | 13.5 | 29.5 | 12.4 | 17.5 | 21.9 | 8.7 | 12.4 | 14.7 | 31.8 | 19.9 | 17.7 | 21.4 | 19.4 | | | |
| ERMViT Vapnik (1999) | NA | NA | NA | NA | NA | NA | NA | NA | NA | NA | NA | NA | NA | NA | NA | NA | NA | NA | 13.5 | 78.7 | 23 |
| DRGen Atwany & Yaqub (2022) | NA | NA | NA | NA | NA | NA | NA | NA | NA | NA | NA | NA | NA | NA | NA | NA | NA | NA | 27.1 | 65.2 | 38.3 |
| SD-ViT Galappaththige et al. (2024) | NA | NA | NA | NA | NA | NA | NA | NA | NA | NA | NA | NA | NA | NA | NA | NA | NA | NA | 34.2 | 71.4 | 46.2 |
| SPSD-ViT Galappaththige et al. (2024) | NA | NA | NA | NA | NA | NA | NA | NA | NA | NA | NA | NA | NA | NA | NA | NA | NA | NA | 34.1 | 77 | 47.3 |
| Mean | 52.1 (29.3) | 51.0 (29.2) | 49.5 (29.8) | 38.9 (22.7) | 19.8 (10.8) | 24.4 (13.6) | 39.4 (15.7) | 24.7 (20.7) | 27.0 (14.4) | 35.5 (22.3) | 14.9 (9.1) | 16.9 (10.4) | 30.7 (17.8) | 17.5 (9.2) | 20.0 (9.1) | 33.8 (21.1) | 19.0 (12.3) | 21.2 (12.2) | 27.2 (9.7) | 73 (6.1) | 38.7 (11.2) |
| Max F1 | | 80.3 (AT) | | | 45.2 (LSTM) | | | 64.3 (GANF) | | | 36 (GAT) | | | 37.8 (GAT) | | | 43 (Omni) | | | 47.3 (SPSD-ViT) | | |
| DCB (%) | | 74% | | | 57% | | | 61% | | | 38% | | | 33% | | | 52% | | | 59% | | |

Table 9: Invariance preservation violation in DR S/M DG (H-scores & DCB). Randomly one class is withheld from the source domain.

| Source | Target | AlexNet H-score | AlexNet DCB | ViT H-score | ViT DCB |
|--------|--------|-----------------|-------------|-------------|---------|
| Messi | A | 17.7 | 13.9 | 35.3 | 38.1 |
| dor-1 | E | 23.2 | 20.0 | 35.7 | 38.1 |
| (M1) | M2 | 52.1 | 34.6 | 54.8 | 52.3 |
| Messi | A | 26.6 | 31.4 | 37.8 | 38.9 |
| dor-2 | E | 34.6 | 46.9 | 42.7 | 42.8 |
| (M2) | M1 | 58.4 | 58.6 | 59.8 | 58.2 |
| Aptos (A) | E | 56.5 | 61.5 | 59.0 | 72.6 |
| Aptos (A) | M1 | 64.6 | 65.2 | 65.0 | 73.8 |
| Aptos (A) | M2 | 63.1 | 72.1 | 62.2 | 66.0 |
| Eye (E) | A | 45.6 | 48.6 | 59.7 | 60.6 |
| Eye (E) | M1 | 35.7 | 45.6 | 52.3 | 55.3 |
| Eye (E) | M2 | 52.0 | 58.6 | 53.4 | 55.3 |
| M1,M2,A | E | 59.0 | 63.5 | 64.8 | 67.0 |
| M1,M2,E | A | 41.5 | 37.7 | 52.9 | 48.4 |
| M1,E,A | M2 | 55.4 | 60.4 | 57.6 | 62.8 |
| M2,E,A | M1 | 51.0 | 57.9 | 52.8 | 59.9 |

Table 10: Performance and correlation with DCB across DomainBed datasets.

| Method | OfficeHome | TerraInc | VLCS | PACS | DomainNet | Corr. w/ DCB |
|--------|-----------|----------|------|------|-----------|--------------|
| SWAD (NeurIPS 2021) | 81.01 | 42.92 | 79.13 | 91.35 | 57.92 | $r = 0.727 \mid p = 0.164$ |
| LP-FT (ICLR 2022) | 81.17 | 47.26 | 80.88 | 92.92 | 57.04 | $r = 0.781 \mid p = 0.119$ |
| WiSE-FT (CVPR 2022) | 86.32 | 54.50 | 82.88 | 97.29 | 58.01 | $r = 0.839 \mid p = 0.076$ |
| MIRO (ECCV 2022) | 84.80 | 59.30 | 82.30 | 96.44 | 60.47 | $r = 0.848 \mid p = 0.070$ |
| DART (CVPR 2023) | 80.93 | 51.24 | 80.38 | 93.43 | 59.32 | $r = 0.781 \mid p = 0.119$ |
| SAGM (CVPR 2023) | 83.40 | 58.64 | 82.05 | 94.31 | 59.05 | $r = 0.863 \mid p = 0.060$ |
| FLYP (CVPR 2023) | 82.76 | 33.25 | 66.64 | 78.53 | 57.41 | $r = 0.577 \mid p = 0.309$ |
| CLIPood (ICML 2023) | 83.31 | 46.28 | 77.19 | 93.16 | 57.78 | $r = 0.752 \mid p = 0.143$ |
| RISE (ICCV 2023) | 78.39 | 49.61 | 80.62 | 93.25 | 55.37 | $r = 0.810 \mid p = 0.097$ |
| VL2V-SD (CVPR 2024) | 87.38 | 58.54 | 83.25 | 96.68 | 62.79 | $r = 0.823 \mid p = 0.087$ |
| DPL (TJSAI 2023) | 84.20 | 52.60 | 84.30 | 97.30 | 59.50 | $r = 0.806 \mid p = 0.100$ |
| SPG (ECCV 2024) | 83.60 | 50.20 | 82.40 | 97.00 | 60.10 | $r = 0.770 \mid p = 0.128$ |
| LPD (ECCV 2024) | 84.21 | 57.30 | 83.20 | 96.55 | 59.30 | $r = 0.846 \mid p = 0.071$ |
| Wen et al. (CVPR 2025) | 87.65 | 63.27 | 84.79 | 97.03 | 63.11 | $r = 0.869 \mid p = 0.056$ |
| DCB | 91.00 | 79.00 | 91.00 | 92.00 | 64.00 | $r = 0.868$ vs Max Accuracy |

(b) For every $\theta_j \in Q_{\text{train}}^{-k}$ compute

$$\rho_{kj} = \frac{\theta_k \cdot \theta_j}{\|\theta_k\| \, \|\theta_j\|}.$$

(c) Average over $j$:

$$\rho_k^{(-k)} = \frac{1}{|Q_{\text{train}}^{-k}|} \sum_j \rho_{kj}.$$

Then compute the overall reference mean:

$$\rho_{\text{avg}} = \frac{1}{|Q_{\text{train}}|} \sum_k \rho_k^{(-k)}.$$

3. **Validation $\rho$-value and deviation.** For each $\theta_m \in Q_{\text{val}}$:

$$\rho_{mj} = \frac{\theta_m \cdot \theta_j}{\|\theta_m\| \, \|\theta_j\|} \quad \forall \theta_j \in Q_{\text{train}},$$

$$\rho_m = \frac{1}{|Q_{\text{train}}|} \sum_j \rho_{mj}, \quad \sigma_m = |\rho_m - \rho_{\text{avg}}|.$$

4. **Conformal interval via order statistic.**
   - Collect the $N_{\text{val}}$ deviations $\{\sigma_m\}$ and sort ascending.
   - Set $\alpha = 0.05$ for a 95% interval.
   - Compute cutoff index
   $$\text{idx} = \left\lfloor \tfrac{N_{\text{val}}+1}{2} \left(1 - \alpha\right) \right\rfloor,$$
   and let $\sigma^*$ be the deviation at that position.
   - The conformal interval is
   $$\text{CI} = [-\sigma^*, +\sigma^*].$$

5. **Implementation notes.**
   - GPU acceleration via CuPy/PyTorch, with CPU fallback.
   - Batching (1 000 images), checkpointing, and garbage collection.
   - Robust error and shape-validation checks.

### J.3 SINGLE-DOMAIN GENERALIZATION (SDG) ANALYSIS

To evaluate how well the learned $\theta$-features generalize across domains, we implemented a statistical domain generalization (SDG). First, $\theta$-vectors and their image IDs are loaded for both the source and target datasets and flattened for analysis. For each target image, we compute the cosine similarity between its $\theta$-vector and all source $\theta$-vectors, yielding a similarity matrix. We then calculate the average similarity ($\rho_{\text{avg}}$) per test image, subtract a precomputed source reference average, and check whether the result falls within the source confidence interval (CI). Each image is labeled "inside" or "outside" the CI, and the final report records the percentage of target images deemed typical of the source distribution.

#### J.3.1 COVERAGE RESULTS

**M1 On**

- **reference_avg** = 0.04047960306748797
- **conf_interval** = $(-0.03824305970024752,\ 0.03824305970024752)$

**M2 On**

- **reference_avg** = 0.041717564380119414
- **conf_interval** = $(-0.043998791641485506,\ 0.043998791641485506)$

**Aptos On**

- **reference_avg** = 0.018978290495836176
- **conf_interval** = $(-0.0596098734708454,\ 0.0596098734708454)$

**EyePACS On**

- **reference_avg** = 0.01288287374567551
- **conf_interval** = $(-0.04568944923551152,\ 0.04568944923551152)$